# TEXT GENERATION WITH EFFICIENT (SOFT) $Q$-LEARNING

## ABSTRACT

Maximum likelihood estimation (MLE) is the predominant algorithm for training text generation models. This paradigm relies on direct supervision examples, which is not applicable to many emerging applications, such as generating adversarial attacks or generating prompts to control language models. Reinforcement learning (RL) on the other hand offers a more flexible solution by allowing users to plug in arbitrary task metrics as reward. Yet previous RL algorithms for text generation, such as policy gradient (*on-policy* RL) and $Q$-learning (*off-policy* RL), are often notoriously inefficient or unstable to train due to the large sequence space and the sparse reward received only at the end of sequences. In this paper, we introduce a new RL formulation for text generation from the soft $Q$-learning (SQL) perspective. It enables us to draw from the latest RL advances, such as path consistency learning, to combine the best of on-/off-policy updates, and learn effectively from sparse reward. We apply the approach to a wide range of text generation tasks, including learning from noisy/negative examples, adversarial attacks, and prompt generation. Experiments show our approach consistently outperforms both task-specialized algorithms and the previous RL methods.

## 1 INTRODUCTION

Recent natural language generation systems have made remarkable progress in producing well-formed coherent text, especially with the massive pretrained language models (LMs) (Radford et al., 2019; Brown et al., 2020; Lewis et al., 2020; Raffel et al., 2019). Those models are typically trained using maximum likelihood estimation (MLE) with a large amount of data supervisions. Despite its successful outcomes, the standard training method suffers from limited applicability to many emerging text generation problems, where little or no supervised data is available. Prominent examples of such low-data problems include generating prompts to control the massive LMs (Yin et al., 2019; Shin et al., 2020; Zhong et al., 2021), learning text generation from noisy or even negative data, generating adversarial text attacks for robustness study (Wallace et al., 2019; Atanasova et al., 2020), and others (Figure 1, right). Due to the failure of standard MLE, people have had to devise specialized algorithms for those problems respectively.

On the other hand, reinforcement learning (RL) (Sutton & Barto, 2018) offers an alternative principled framework for learning from arbitrary reward functions, and has achieved great advances in robotic and game control. However, RL by far has made limited success for training text generation, primarily due to the key challenges of *sparse reward* (i.e., a single reward signal is received only after the whole text sequence is generated) and *large action space* (i.e., a vocabulary of millions of words). For instance, a popular family of RL algorithms studied extensively for text generation is the policy-based (Williams, 1992) or actor-critic based (Bahdanau et al., 2016; Rennie et al., 2017) algorithms, with policy gradient (PG) being the most prevalent example (Ranzato et al., 2016; Li et al., 2016; Rennie et al., 2017; Tan et al., 2018; Pasunuru & Bansal, 2018; Paulus et al., 2018). Those algorithms train the model with *on-policy* updates, i.e., the text samples used for estimating policy gradients are from the target model itself. Due to the exponentially large space of sequences, on-policy updates often suffer from extremely high variance and low data efficiency (e.g., most model samples are not useful for learning). Thus directly training with PG from scratch is usually impossible. In practice, the model has to be initialized by MLE training, followed by PG as finetuning, which often leads to limited improvement (Choshen et al., 2020; Wu et al., 2018).

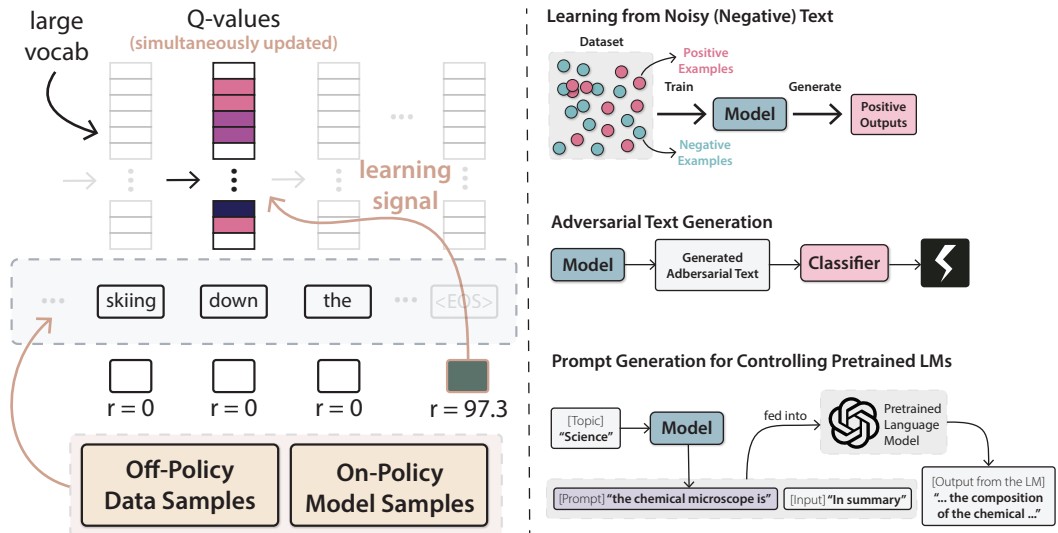

Figure 1: **Left:** An overview of the proposed SQL algorithm for text generation. Text generation is challenging due to sparse reward (i.e., the rewards of all intermediate steps are 0) and large action space (i.e., large vocabulary). Our SQL formulation enables several key algorithmic features as highlighted with yellow color, including (1) the combined on- and off-policy updates for the best of both, (2) bridging the final non-zero reward to directly supervise the $Q$-value estimation at intermediate steps for learning stability, and (3) simultaneously updating the $Q$-values of all candidate actions for efficiency. **Right:** We explore diverse applications of the text-generation RL algorithm.

Another set of work has resorted to *off-policy* RL. The key advantage is that samples from other sources, e.g., human-written text, can be used, making them more data efficient than on-policy methods. Previous work has used either importance weighted PG (Pang & He, 2021; Zhou et al., 2017; Kandasamy et al., 2017) or $Q$-learning based algorithms (Guo, 2015; Jaques et al., 2020; Narasimhan et al., 2015). However, off-policy methods have been considered to be less stable. For example, the $Q$-learning performance relies heavily on how accurate the learned $Q$-function assesses the quality of intermediate subsequences – a challenging task due to the sparse reward signals.

In this paper, we develop a new RL formulation for text generation that tackles the above issues (Figure 1, left). We reframe the text generation problem from the *soft Q-learning* perspective originally developed in robotics (Haarnoja et al., 2017; Schulman et al., 2017). The resulting connection allows us to seamlessly take advantage of the latest successful techniques from the RL literature. In particular, we introduce and adapt the principled *path consistency learning* (Nachum et al., 2017) to text generation, that (1) offers a natural way to train the model with both on- and off-policy updates, hence combining the best of the two strategies, (2) bridges the sparse reward signal to directly supervise the $Q$ function learning, leading to more accurate $Q$ estimation and credit assignment, and (3) makes efficient updates to $Q$-values by considering all candidate actions together.

The generality and efficiency of the proposed method allows us to train text generation in a wide range of applications: (1) With *noisy and negative training examples*, our approach learns to generate accurate entailment text that greatly improves upon the data itself as well as other various training methods; (2) Our approach also manages to train an effective *adversarial text generator* for robustness test for classifiers; (3) We train a *prompt generator* with our algorithm to achieve controllable generation of pretrained LMs in terms of topics. On all the three tasks, our approach consistently improves over not only previous RL algorithms for text generation, but also diverse task-specialized methods designed specifically for each of the problems, respectively. In the appendix (§A.1.4), we also show that on standard supervised tasks where MLE prevails, our approach is competitive to train text generation models *from scratch*, which was usually impossible for previous RL algorithms.

## 2 BACKGROUND AND CHALLENGES

The goal of text generation is to produce coherent text $\boldsymbol{y} = (y_0, ..., y_T)$ of certain properties for a given task, where $y_t$ is a token from a vocabulary $\mathcal{V}$, and $T$ is the text length. The generation can condition on arbitrary input context, which we omit for simplicity of notations. We aim to learn a

generation model $p_\theta(\boldsymbol{y})$ which is typically decomposed autoregressively as $p_\theta(\boldsymbol{y}) = \prod_{t=0}^{T} p_\theta(y_t \mid \boldsymbol{y}_{<t})$, where $\boldsymbol{y}_{<t} = (y_0, ..., y_{t-1})$ is the prefix, and the distribution at each step $t$ is obtained by applying the softmax function on the output logits:

$$p_\theta(y_t \mid \boldsymbol{y}_{<t}) = \frac{\exp f_\theta(y_t \mid \boldsymbol{y}_{<t})}{\sum_{y' \in \mathcal{V}} \exp f_\theta(y' \mid \boldsymbol{y}_{<t})}. \tag{1}$$

Here $f_\theta(y \mid \boldsymbol{y}_{<t})$ is the logit of token $y$ computed by the generation model.

Given a training example $\boldsymbol{y}^*$, maximum likelihood training (MLE) updates the model with the gradient $\nabla_\theta \mathcal{L}_{\text{MLE}}(\boldsymbol{\theta}) = \sum_{t=0}^{T} \nabla_\theta \log p_\theta(y_t^* \mid \boldsymbol{y}_{<t}^*)$. Despite its popularity, MLE-based training only applies when clean supervised data $\boldsymbol{y}^*$ is available, and cannot be used to optimize arbitrary task metrics (e.g., BLEU, entailment score) which are typically the goal in many text generation tasks.

## 2.1 REINFORCEMENT LEARNING (RL) FORMULATIONS FOR TEXT GENERATION

**Notations.** Previous research has formulated text generation as an RL problem by considering the following finite-time Markov Decision Process (MDP). At each time step $t$, let the "state" be $\boldsymbol{s}_t = \boldsymbol{y}_{<t}$, namely the partial sequence generated so far. The model, also known as the "agent", takes as input the current state $\boldsymbol{s}_t$ and outputs a token, also called "action", $a_t \in \mathcal{V}$ according to a policy $\pi(a_t \mid \boldsymbol{s}_t)$. The agent then receives a reward $r_t = r(\boldsymbol{s}_t, a_t)$ and deterministically transitions to next state $\boldsymbol{s}_{t+1}$ (i.e., the concatenation of the tokens in $\boldsymbol{s}_t$ and the new token $a_t$).

Following the notation convention in RL, let $\tau$ be the trajectory (i.e., text sample) generated by the policy. The agent's objective is to maximize the accumulative reward, $J(\pi) = \mathbb{E}_{\tau \sim \pi} \left[ \sum_{t=0}^{T} \gamma^t r_t \right]$, where $\gamma \in (0, 1]$ is the discount factor. A central concept in RL is the $Q$-function of policy $\pi$, defined as $Q^\pi(\boldsymbol{s}_t, a_t) = \mathbb{E}_\pi \left[ \sum_{t'=t}^{T} \gamma^{t'} r_{t'} \mid \boldsymbol{s}_t, a_t \right]$, which is the expected future reward of taking action $a_t$ (i.e., generating token $a_t$) in state $\boldsymbol{s}_t$ and continuing with the policy $\pi$.

**Challenges.** Text generation poses significant challenges to RL, particularly because (1) the reward signal is usually sparse, i.e., $r_t = 0$, $\forall t < T$ and the agent receives a non-zero reward $r_T$ only after it generates the full sequence, (2) the action space (i.e., the vocabulary $\mathcal{V}$) is extremely large, often containing millions of words. The challenges have led to difficulties of the two major families of RL approaches applied to text generation problems, as detailed below.

**Policy-based RL** techniques directly parameterize the policy $\pi_\theta$ with parameters $\boldsymbol{\theta}$. Thus the policy $\pi_\theta(a_t \mid \boldsymbol{s}_t)$ exactly corresponds to the above generation model $p_\theta(y_t \mid \boldsymbol{y}_{<t})$. *Policy gradient (PG)* is one of the most widely used algorithms for text generation (Ranzato et al., 2016). It optimizes the cumulative reward with the policy gradient:

$$\nabla_\theta J(\pi_\theta) = -\mathbb{E}_{\tau \sim \pi_\theta} \left[ \sum_{t=0}^{T} \hat{Q}(\boldsymbol{s}_t, a_t) \nabla_\theta \log \pi_\theta(a_t \mid \boldsymbol{s}_t) \right], \tag{2}$$

where $\hat{Q}(\boldsymbol{s}_t, a_t)$ is the estimated $Q^{\pi_\theta}$ value with sample $\tau$. Notice that the expectation is taken w.r.t. the policy $\pi_\theta$, which makes PG an *on-policy* algorithm, meaning that the sample $\tau$ needs to come from the the current policy $\pi_\theta$ itself. In practice, however, optimizing this objective alone from scratch is unlikely going to work because most samples $\tau \sim \pi_\theta$ are just gibberish with zero reward, failing to provide meaningful training signals for updating the policy. Previous literature either initializes the policy $\pi_\theta$ with MLE training, and/or use a combination of MLE and PG updates, which often leads to marginal gains in practice (Wu et al., 2018; Choshen et al., 2020).

**Value-based RL** techniques, such as $Q$-*learning*, implicitly learn the policy $\pi$ by approximating the value $Q^\pi(\boldsymbol{s}, a)$ directly. Specifically, let $Q^*(\boldsymbol{s}, a) = \max_\pi Q^\pi(\boldsymbol{s}, a)$ denote the optimal value over policies. Thus the optimal policy $\pi^*$ is simply taking the action of maximal $Q^*$ value at each state. The approximation of $Q^*$ is based on the well-known Bellman temporal consistency:

$$Q^*(\boldsymbol{s}_t, a_t) = r_t + \gamma \max_{a_{t+1}} Q^*(\boldsymbol{s}_{t+1}, a_{t+1}). \tag{3}$$

Deep $Q$-learning (Mnih et al., 2013) parameterizes the $Q$-function as $Q_\theta(\boldsymbol{x}, a)$ (e.g., a neural network), and train the parameters by minimizing the following regression objective:

$$\mathcal{L}(\boldsymbol{\theta}) = \mathbb{E}_{\pi'} \left[ 0.5 \cdot \left( r_t + \gamma \max_{a_{t+1}} Q_{\bar{\theta}}(\boldsymbol{s}_{t+1}, a_{t+1}) - Q_\theta(\boldsymbol{s}_t, a_t) \right)^2 \right], \tag{4}$$

$$\mathcal{L}_{\text{SQL, PCL}}(\boldsymbol{\theta}) = \mathbb{E}_{\pi'}\left[\frac{1}{2}\left(-V_{\bar{\theta}}(\boldsymbol{s}_t) + \gamma V_{\bar{\theta}}(\boldsymbol{s}_{t+1}) + r_t - \log \pi_{\theta}(a_t \mid \boldsymbol{s}_t)\right)^2\right] \quad \mathcal{L}_{\text{SQL, PCL-ms}}(\boldsymbol{\theta}) = \mathbb{E}_{\pi'}\left[\frac{1}{2}\left(-V_{\bar{\theta}}(\boldsymbol{s}_t) + \gamma^{T-t} r_T - \sum_{l=0}^{T-t} \gamma^l \log \pi_{\theta}(a_{t+l} \mid \boldsymbol{s}_{t+l})\right)^2\right]$$

Figure 2: Soft $Q$-Learning with path consistency learning (PCL) objectives, where we illustrate with a vocabulary of size 3. **Left:** Single-step objective (Eq.9), where for each $(\boldsymbol{s}_t, a_t)$, the computation involves step $t$ and $t+1$. Dashed boxes in **dark green** and **gray** indicate the regression target, where the intermediate reward $r_t$ is often 0 due to sparsity. The gradient is applied to parameters $\boldsymbol{\theta}$ at step $t$ (indicated by **orange** color). **Right:** Multi-step objective (Eq.11) which aggregates from step $t$ all the way to $T$. In this way, the final-step non-zero reward $r_T$ is used as the regression target.

where $\bar{\boldsymbol{\theta}}$ is the parameters of the *target* $Q$-network, which is a slow copy of $\boldsymbol{\theta}$ and considered as constant for gradient computation of $\boldsymbol{\theta}$. Here $\pi'$ is an *behavior policy* which can be an arbitrary distribution over text, such as the data distribution or replay buffer (Mnih et al., 2013). This makes $Q$-learning an *off-policy* algorithm because of its ability to use samples coming from other policies. After learning $Q_{\theta}$, one can induce a policy $\pi$ from it that takes $\arg\max_a Q_{\theta}(\boldsymbol{s}, a)$ at each state $\boldsymbol{s}$. Jaques et al. (2017) instead sample tokens from the softmax function applied to $Q_{\theta}$.

However, the training can be unstable and inefficient due to several challenges: **(1)** The bootstrapping nature of the above regression problem can make the training unstable. That is, the regression target $r_t + \gamma \max_{a_{t+1}} Q_{\bar{\theta}}(\boldsymbol{s}_{t+1}, a_{t+1})$ itself is derived from the $Q$-function to be learned (Kumar et al., 2019). The problem is exacerbated in the presence of sparse reward in text generation, where the real observed signal $r_t$ is zero for all intermediate $t < T$; **(2)** The large action space (e.g., $10^4$) in text generation results in slow updates. In particular, notice that Eq.(4) applies the gradient update to the $Q_{\theta}$-value of the *only one* particular token $a_t$ (out of the $10^4$ candidate tokens in the vocabulary), making the training inefficient; **(3)** Besides, pure off-policy updates could be highly sensitive to the quality of training data, and miss the opportunity of on-policy exploration that maximizes the reward of interest in a more direct way.

## 3 THE SOFT $Q$-LEARNING FRAMEWORK

In this section, we combat the difficulties of previous RL methods by introducing the soft $Q$-learning (SQL) formulation of text generation. We show that the formulation is seamlessly compatible with the common architecture of text generation model (Eq.1), permitting easy implementation (§3.1). The formulation further allows us to integrate the latest advances in RL, notably path consistency learning (Nachum et al., 2017) that makes the RL training efficient and stable in practice (§3.2). Figure 2 and Algorithm 1 summarizes the resulting SQL framework for efficient training.

### 3.1 SOFT $Q$-LEARNING FORMULATION FOR TEXT GENERATION

Soft $Q$-learning (Haarnoja et al., 2017; Schulman et al., 2017; Nachum et al., 2017) is an maximum-entropy (MaxEnt) extension to the standard (hard) $Q$-learning (Mnih et al., 2015; Sutton & Barto, 2018). Under this framework, the agent is encouraged to optimize the reward while staying as stochastic as possible, with the objective $J_{\text{MaxEnt}}(\pi) = \mathbb{E}_{\tau \sim \pi}\left[\sum_{t=0}^{T} \gamma^t r_t + \alpha \mathcal{H}(\pi(\cdot \mid \boldsymbol{s}_t))\right]$, which augments the vanilla $J(\pi)$ with the additional Shannon entropy term $\mathcal{H}$ with coefficient $\alpha$.[1] This is appealing because it seamlessly connects the $Q$-values to the familiar output *logits* of a text generation model, which enables straightforward implementation of the SQL formulation.

$Q$-**values as Generation Model Logits.** We show the connection of the $Q$-values with the logits, i.e., the model outputs right before the softmax layer. Concretely, with the SQL objective, the following relationship between optimal policy $\pi^*$ and action-value $Q^*$ holds (Haarnoja et al., 2017; Schulman et al., 2017):

$$\pi^*(a \mid \boldsymbol{s}) = \frac{\exp Q^*(\boldsymbol{s}, a)}{\sum_{a'} \exp Q^*(\boldsymbol{s}, a')}. \tag{5}$$

---

[1] WLOG, we can assume $\alpha = 1$, as it can be folded into the reward function by scaling the latter with $1/\alpha$.

This form is highly reminiscent of the $\mathrm{softmax}$ layer of the generation model in Eq.(1). The connection suggests that we can naturally parameterize the $Q$-function in SQL as the generation model logit function, i.e., $Q_\theta(s, a) \equiv f_\theta(a \mid s)$. In other words, *the model output $f_\theta(a \mid s)$, originally interpretted as the "logit" of token $a$ given the preceding tokens $s$, is now re-interpreted as the $Q$-value of action $a$ in state $s$.* When achieving optimality, $f_{\theta^*}(a \mid s)$, namely $Q^*(s, a)$, represents the best possible future reward achievable by generating token $a$ in state $s$. Similarly, the full generation model $p_\theta(a \mid s)$ in Eq.(1) that applies $\mathrm{softmax}$ to $f_\theta$ now precisely corresponds to the policy $\pi_\theta$ induced from $Q_\theta(s, a)$. That is,

$$\pi_\theta(a \mid s) = \frac{\exp Q_\theta(s, a)}{\sum_{a'} \exp Q_\theta(s, a')} \equiv \frac{\exp f_\theta(a \mid s)}{\sum_{a'} \exp f_\theta(a' \mid s)} = p_\theta(a \mid s). \tag{6}$$

We could further gain even more intuitive interpretation of the above generation policy $\pi^*$ from the lens of *advantage* function (Sutton & Barto, 2018). Specifically, in SQL, the optimal *state-value* function is the log-normalizer of the optimal $Q$-values (Haarnoja et al., 2017; Schulman et al., 2017). This allows us to rewrite Eq.(5) into a more concise form:

$$V^*(s) = \log \sum_{a'} \exp Q^*(s, a'), \quad \pi^*(a \mid s) = \exp\left(Q^*(s, a) - V^*(s)\right) = \exp A^*(s, a), \tag{7}$$

where $A^*$ is the optimal advantage function. The equation says that, in the proposed text generation SQL formulation, the optimal policy generates token $a$ in state $s$ according to the token's advantage.

## 3.2 EFFICIENT TRAINING WITH PATH CONSISTENCY

The above section has described parameterizing the $Q$-function with the common generation model with parameters $\theta$. Now we present how to learn the $Q_\theta$ function within the SQL framework. Vanilla training based on the Bellman temporal consistency can suffer from the instability and inefficiency issues similar to the conventional $Q$-learning (§2.1), as we discuss more in the appendix (§A.3.2). Fortunately, our SQL formulation allows us to import latest advances of RL techniques to the text generation setting that overcome the difficulties.

Specifically, we adapt the *unified path consistency learning (PCL)* that has excelled in game control (Nachum et al., 2017). The PCL-based training updates $Q$-values of *all* tokens at once through a connection between the value function and the induced policy. More specifically, it is shown in Nachum et al. (2017) that the optimal policy $\pi^*$ (Eq.5) and the optimal state value function $V^*$ (Eq.7) in SQL must satisfy the following consistency property for all states and actions:

$$V^*(s_t) - \gamma V^*(s_{t+1}) = r_t - \log \pi^*(a_t \mid s_t), \quad \forall s_t, a_t. \tag{8}$$

Accordingly, the PCL-based training attempts to encourage the satisfaction of the consistency with the following regression objective:

$$\mathcal{L}_{\mathrm{SQL, PCL}}(\theta) = \mathbb{E}_{\pi'}\left[\frac{1}{2}\left(-V_{\bar\theta}(s_t) + \gamma V_{\bar\theta}(s_{t+1}) + r_t - \log \pi_\theta(a_t \mid s_t)\right)^2\right], \tag{9}$$

where $\pi_\theta$ is the induced policy defined in Eq.(6); $V_{\bar\theta}$ is defined similarly as in Eq.(7) but depends on the target $Q_{\bar\theta}$ network (i.e., a slow copy of the $Q_\theta$ to be learned), and recall that $\pi'$ is an arbitrary behavior policy (e.g., data distribution). Please see Figure 2 (left) for an illustration. Crucially, notice that the gradient update is applied to $\theta$ through the $\log \pi_\theta$ term which explicitly involves the $Q_\theta$-values of *all* tokens $a$ in the vocabulary. This shows an important difference from the above vanilla training in conventional $Q$-learning (§2.1) where $Q_\theta$ is updated only through the particular $a_t$ token. The PCL training thus offers more efficient updates for the $Q_\theta$ function.

**Multi-step PCL for Sparse Reward.** The above PCL objective Eq.(9) alone does not resolve the potential instability issue due to the bootstrapped $V_{\bar\theta}(s_{t+1})$ value and the sparse reward (i.e., $r(s_t, a_t) = 0$ for $t < T$). Our SQL formulation allows us to additionally incorporate the *multi-step* variant of the PCL training (Nachum et al., 2017) to resolve the issue. Specifically, by applying a telescoping sum on the consistency equation (Eq.8) starting from $t$ up to $T$, we arrive at the multi-step temporal consistency:

$$V^*(s_t) - \gamma^{T-t} V^*(s_{T+1}) = \sum_{l=0}^{T-t} \gamma^l \left(r_{t+l} - \log \pi^*(a_{t+l} \mid s_{t+l})\right), \tag{10}$$

where the value of past-terminal state is zero, $V^*(s_{T+1}) = 0$; and the rewards are only available at the end, $\sum_{l=0}^{T-t} \gamma^l r_{t+l} = \gamma^{T-t} r_T$. We can then come to the following multi-step objective function,

$$\mathcal{L}_{\text{SQL, PCL-ms}}(\boldsymbol{\theta}) = \mathbb{E}_{\pi'} \left[ \frac{1}{2} \left( -V_{\bar{\theta}}(s_t) + \gamma^{T-t} r_T - \sum_{l=0}^{T-t} \gamma^l \log \pi_\theta (a_{t+l} \mid s_{t+l}) \right)^2 \right]. \tag{11}$$

We can see the objective side-steps the need to bootstrap intermediate value functions $V_{\bar{\theta}}(s_{t'})$ for $t' > t$. Instead, it directly uses the non-zero end reward $r_T$ to derive the update for $\boldsymbol{\theta}$. Please see Figure 2 (right) for an illustration. In practice, we combine the single- and multi-step objectives (Eqs.9 and 11) together for training.

**Joint On- and Off-policy Training.** Finally, we highlight that the behavior policy $\pi'$ involved in the objectives Eqs.(9) and (11) can be an arbitrary policy (i.e., distribution over text sequences), from which we can draw trajectories $\tau$ (i.e., text samples). For example, $\pi'$ can be a (possibly noisy) text dataset, or a set of text samples produced by other generation models, resulting in off-policy training. We can also set $\pi'$ to be the current generation model $\pi_\theta$ to be learned, resulting in on-policy training. In practice, we could first train the model with only off-policy data for warming up, and then continue with joint on- and off-policy training to further maximize the reward.

---

**Algorithm 1** Efficient Soft $Q$-Learning for Text Generation

---

**Input:** $Q_\theta$ function (i.e., generation model logit function $f_\theta$ in Eq.1)
         Reward function $r(s, t)$
         Training examples $\mathcal{D}$ (for off-policy updates; *optional*)
 1: Initialize $\boldsymbol{\theta}$ and target model parameters $\bar{\boldsymbol{\theta}}$
 2: **repeat**
 3:     Draw a batch of off-policy samples $\{\tau_{\text{off}}\} \sim \mathcal{D}$
 4:     Draw a batch of on-policy samples $\{\tau_{\text{on}}\}$ by decoding with policy $\pi_\theta(a_t \mid s_t)$ (Eq.6)
 5:     Compute $Q_\theta(s_t, a_t)$ values (i.e., the model logits) and target $Q_{\bar{\theta}}(s_t, a_t)$ for $(s_t, a_t) \in \{\tau_{\text{off}}\} \cup \{\tau_{\text{on}}\}$
 6:     Compute the objectives in Eqs.(9) and (11)
 7:     Update the model parameters $\boldsymbol{\theta}$ via gradient descent
 8:     Update the target model parameters $\bar{\boldsymbol{\theta}}$ by $\bar{\boldsymbol{\theta}} \leftarrow \rho \bar{\boldsymbol{\theta}} + (1 - \rho) \boldsymbol{\theta}$ with update rate $\rho$
 9: **until** convergence
**Output:** The trained $Q_{\theta^*}$ function and the induced generator $\pi_{\theta^*}$

---

## 4    Applications and Experiments

### 4.1    Learning from Noisy (Negative) Text

The popular MLE algorithm learns by (blindly) imitating training data. However, it is often expensive to curate clean quality data. It is thus highly desirable to be able to learn from data with noises, or even *negative* examples. With the guidance of task metrics (rewards), the model can even learn to "outperform" the training data and achieve desired generation behaviors. To this end, we consider the task of *entailment generation* (Pasunuru & Bansal, 2017). Given a sentence (premise), the goal is to generate a new sentence (hypothesis) that logically follows the premise. For example, given source sentence "`Sophie is walking a dog outside her house`", the hypotheses "`Sophie is outdoor`" is considered entailed, but "`Sophie is inside her house`" is not and even is a negative (contradictive) sentence.

**Setup (more in the appendix §A.2.1).** We sub-sampled $50k$ training examples from the SNLI dataset (Bowman et al., 2015), a commonly used entailment classification dataset. The hypotheses have an average entailment probability of only $50\%$, and over $2/5$ of them less than $20\%$ (negative/contradictive examples). This poses a significant challenge for the models to learn from the noises. The rewards used in RL algorithms include (1) the entailment score of the generation measured by a robust entailment classifier (Nie et al., 2020), (2) the log-likelihood of the generation as an indicator of language quality measured by a GPT-2 language model (Radford et al., 2019), and (3) BLEU score w.r.t the input premises as another language quality reward that avoids trivial outputs. We sum together all rewards with weights 1.0.

We compare our approach with a broad range of baselines, including (1) the standard MLE training (`MLE`); (2) `MLE+reward`, where we use the reward function to filter examples; (3) joint MLE and

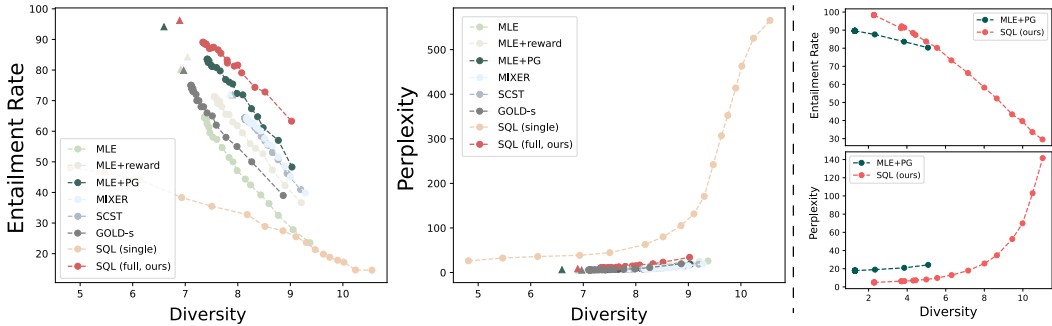

Figure 3: **Left:** entailment generation performance plotted against diversity (average of $H_1$ and $H_2$). Circles represent results of top-$p$ sample outputs, and triangles represent results of beam-search outputs. **Right:** entailment attack performance against diversity (average of $H_1$ and $H_2$). Only a few `MLE+PG` dots are visible because the model is not able to generate more diverse samples even with increasing $p$ value in top-$p$ decoding, i.e., the model collapses.

PG training with MLE initialization (`MLE+PG`), where we initialize the model with MLE training, then train it with combined MLE and PG losses; previous text-generation RL algorithms including (4) `MIXER` (Ranzato et al., 2016), (5) `Self-critic` (Rennie et al., 2017), and (6) one of the latest methods `GOLD-s` (Pang & He, 2021) which is a pure off-policy method based on importance-sampling PG. To ablate the effect of multi-step training (§3.2), we additionally compare with a simplified variant of our approach that uses only vanilla single-step PCL training (`SQL(single)`). In the appendix (§A.1.1) we compare and discuss more baselines such as MLE weighted by rewards.

We evaluate generation results in terms of entailment rate, language quality (perplexity), and diversity which is measured by the Shannon entropy over unigrams and bigrams ($H_1$, $H_2$) (Gehrmann et al., 2021). Since text generation models intrinsically trade off diversity and quality (Caccia et al., 2019; Hashimoto et al., 2019), we vary the generation diversity by generating samples via top-$p$ sampling (Holtzman et al., 2019) with different $p$ values, and plot the entailment rate and perplexity against diversity, resp. We also evaluate the samples produced by beam-search decoding.

**Results.** Figure 3 (left) shows the results. First, notice that `MLE` performs poorly, while `MLE+reward` improves upon it. This is not surprising as the training data contain noisy/negative examples. Similarly, since the pure off-policy algorithm `GOLD-s` relies heavily on the data distribution, we observed that it achieves sub-optimal performance. The on-policy `MLE+PG` with MLE initialization gives better entailment rate. In comparison, our full `SQL` framework achieves the best entailment-diversity trade-off. The comparison between `SQL` and `SQL(single)` highlights the importance of having the multi-step objective which directly uses the end reward rather than bootstrapping intermediate $Q$-values for supervision.

### 4.2 *Universal* ADVERSARIAL ATTACKS

We next study the application in text adversarial attacks, where again no supervised data is available. Adversarial attacks is an increasingly important research topic as they reveal models' vulnerabilities and flaws. This is especially true for universal attacks (Wallace et al., 2019; Atanasova et al., 2020), where we want to generate universal examples that trick the model on *all* possible inputs. For instance, consider the context of entailment classification. Our goal is to find universal human-readable hypotheses that are going to be classified as "entailment" with as high probability as possible, regardless of the input premises. This is a more challenging setting compared to previous instance-specific attack (Morris et al., 2020; Jin et al., 2020; Ebrahimi et al., 2017) where the attack model conditions on a premise and generates an adversarial hypothesis specific to the premise.

**Setup (more in the appendix §A.2.2).** We aim to attack one of the most popular MultiNLI (Williams et al., 2018) entailment classifiers on HuggingFaceHub.[2] The attack generation model generates adversarial text without conditioning on any inputs so that the generated attacks are universal to all premises. We compare our `SQL` with `MLE+PG`. We use all hypotheses in the MultiNLI dataset as the training data for the MLE training in `MLE+PG` and the off-policy updates for our `SQL`. We do not compare with previous specialized adversarial text attack methods, because they either are not applicable to the challenging universal attack setting (Morris et al., 2020;

---

[2]https://github.com/pytorch/fairseq/tree/master/examples/roberta

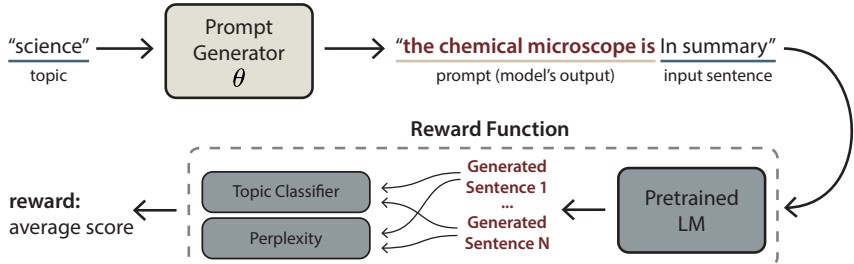

Figure 4: Conditioning on a topic (e.g., ``science''), the prompt generator automatically produces a short piece of text (i.e., prompt) such that, by prepending the prompt to the input text, the pretrained LM will generate continuation sentences of the particular topic. The subsequent components serve as the reward functions to train the prompt generator. The discrete steps (highlighted in **red**) make previous gradient-based prompt tuning approaches not applicable here.

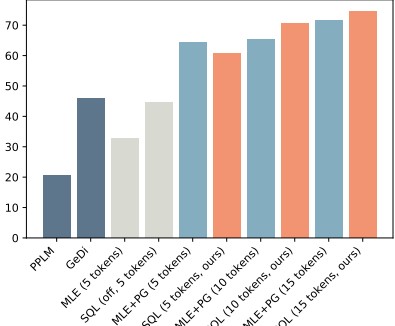

Figure 5: Average topic accuracy.

| PPLM | GeDi | MLE (5) | SQL (off, 5) |
|---|---|---|---|
| 13.07 | 123.88 | 25.70 | 25.77 |

| MLE+PG (5/10/15) | | SQL (5/10/15, ours) | |
|---|---|---|---|
| 25.52/28.16/28.71 | | 25.94/26.95/29.10 | |

Table 1: Language perplexity results averaged across topics. The lower, the more fluent the generated continuation sentences.

| Model | PPLM | GeDi | SQL |
|---|---|---|---|
| Seconds | 5.58 | 1.05 | 0.07 |

Table 2: Average sentence generation time cost.

Jin et al., 2020; Ebrahimi et al., 2017), or were not designed to generate human-readable sentences (Wallace et al., 2019). We use similar settings as in §4.1 to explore the diversity-quality trade-off by plotting the entailment rate and perplexity against diversity, respectively. The entailment classifier to be attacked is used as entailment score reward functions. We additionally include a token-level repetition penalty reward for readability.

**Results.** Figure 3 (right) shows the results, and Table 4 (appendix) shows samples. We can see that `SQL` outperforms `MLE+PG` consistently across different diversity values. The outputs from `MLE+PG` are not diverse even with high $p$'s, indicating the model collapses and can only generate a small set of unique adversarial examples. The model by `SQL` discovers the pattern "saint-pierre-et-saint-paul" (an entity name), and exploits this to generate samples with high universal entailment rate.

## 4.3 PROMPT GENERATION FOR CONTROLLING PRETRAINED LANGUAGE MODELS

A reward function does not just have to be a metric like the BLEU score, but also a complicated pipeline that eventually returns a score. To demonstrate this, we consider the emerging task of prompting a large pretrained LM for controllable generation (Hu et al., 2017; Radford et al., 2019; Brown et al., 2020). The goal is to learn to generate text prompts that steer the LM to generate sentences of certain desired attributes (e.g., topics). The problem of controlling the generation of pretrained LMs was previously approached through specialized algorithms such as modifying the LM hidden states during decoding (Dathathri et al., 2020; Krause et al., 2020; Qin et al., 2020). Here we show that prompts offer an easier, faster, more effective way for controlled generation.

Learning to generate/tune prompts is gaining increasing attention recently. It side-steps the needs for expensive LM fine-tuning, and adapts LMs to new scenarios with prompt as the (compute-friendly) interface. Most existing approaches (Wallace et al., 2019; Li & Liang, 2021; Lester et al., 2021) rely on gradient backpropagation and are applicable only when the whole training pipeline is differentiable. This does not hold for the text generation setting, as illustrated in Figure 4. In contrast, the RL framework is generally applicable to any differentiable or discrete pipelines.

**Setup (more in the appendix §A.2.3).** Following (Dathathri et al., 2019), we aim to control the generation to have one of 7 topics (e.g., "science"); the generated prompt is prepended to one of 20 input sentences for the pretrained LM to generate continuation sentences. Figure 4 shows the

architecture of prompt-based controllable generation. We compare our `SQL` method with `MLE+PG` as before. Since the prompt length could impact the generated sentences, we conducted experiments with maximum prompt length 5, 10, and 15. As ablation study, we also evaluate the SQL algorithm with only off-policy updates (i.e., without on-policy exploration), denoted as `SQL(off)`, and compare it with vanilla `MLE` training. Finally, we also compare with two specialized controllable generation techniques based on pretrained LMs, namely `PPLM` (Dathathri et al., 2019) and `GeDi` (Krause et al., 2020), following similar procedures using their open-sourced code. We use a distilled GPT-2 model[3] as the pretrained LM to be controlled. For rewards, we use the topic accuracy of the continuation sentences measured by a *zero-shot* classifier, plus the the log-likelihood of continuation sentences as the language quality reward measured by a distilled GPT-2.

**Results** Figure 5 shows the topic accuracy of the controlled LM outputs averaged across the 7 topics, and Table 1 shows the respective language quality results. More detailed topic accuracy results and samples are provided in the appendix (§A.1.3) (where `GeDi` obtained low accuracy on 2 of the 7 topics, possibly because the topic tokens are tokenized into two subwords for which the model released by the authors was not specifically trained). We can see that the prompts generated by our `SQL` cause the LM to generate sentences with high topic accuracy while maintaining low perplexity in most settings. Increasing the prompt length positively impacts the topic accuracy, which makes sense because longer prompts give more flexible for steering the LM. The comparison between `MLE` and `SQL(off)` shows that the off-policy component of SQL is better than standard MLE training, as it incorporates reward signals instead of just blindly following the (noisy) data.

Next, comparing with the previous steered decoding such as `PPLM` and `GeDi`, we can see the prompt-based control trained with RL achieves better trade-off between topic accuracy and language quality. Moreover, once a prompt is produced, we can use the pretrained LM to generate text of desired topics efficiently, with the same time cost as standard non-controlled decoding. In comparison, the dedicated steered decoding is often orders-of-magnitude slower, as shown in Table 2.

## 5 RELATED WORK

Standard RL algorithms maximizing the external rewards can sometimes be over-sensitive to the randomness in the environment. Recent works have considered maximum-entropy RL (MaxEnt RL) extensions, such as the soft $Q$-learning (SQL) (Haarnoja et al., 2017; Nachum et al., 2017; Schulman et al., 2017), that maximize the entropy of policy besides the rewards, and have demonstrated substantial improvement in robotic and game control (Ziebart et al., 2008; O'Donoghue et al., 2017; Nachum et al., 2018; Eysenbach & Levine, 2021). Our work is the first to adapt SQL and its advanced variants (in particular the path consistency learning (Nachum et al., 2017)) to the challenging text generation problem and show significant results on diverse applications. Applying RL for text generation has been discussed in alleviating the exposure bias problem and optimizing task metrics (Li et al., 2016; Wu et al., 2016; Rennie et al., 2017; Paulus et al., 2018; Chen & Bansal, 2018). For example, Ranzato et al. (2016) used the REINFORCE algorithm (Williams, 1992), and Bahdanau et al. (2016) used the actor-critic algorithm; Guo et al. (2018) and Shi et al. (2018) tried to relieve the sparsity problem via hierarchical and inverse RL methods, resp. They are all on-policy RL algorithms with the need of pretraining their models using MLE. Another line of work focused mostly on using only off-policy data, often for offline training of chatbots (Jaques et al., 2020; Kandasamy et al., 2017; Zhou et al., 2017; Pang & He, 2021). As a result, the opportunity of directly improving the reward (as in on-policy updates) for other rich tasks is missed. Our proposed framework combines on- and off-policy training, and further offers solutions for efficient training from scratch in the presence of large action space and sparse sequence-level reward in text generation.

## 6 CONCLUSION

We develop a new RL formulation for text generation based on soft $Q$-learning and path consistency learning. We conduct experiments on learning with noisy and negative data, black box adversarial attack, prompting a pretrained language model for controllable generation, and finally, on standard supervised tasks. The RL formulation opens up enormous new opportunities to integrate more advances made in the fertile RL literature to improve text and other sequence generation problems.

---

[3]`https://huggingface.co/distilgpt2`

## 7 ETHICS STATEMENT

This work develops a new RL formulation for text generation. While we demonstrate the framework in four applications, it could be adapted to other (emerging) applications. One major component in these applications is the design of the reward function, which influences the behavior of the trained agent. While we believe the MaxEnt RL framework is more robust against reward misspecification (Eysenbach & Levine, 2021), the potential failures of sub-optimal reward functions are widely known and discussed.[4] To this end, deploying this model to the wild requires careful and extensive examination, using tools such as Ribeiro et al. (2020). Further, we highlight the application for (black-box) adversarial attacks in the paper, with the intention of using adversarial attacks to understand the model's inner workings. That being said, this could potentially be misused to conduct malicious attacks against systems. Hence, users of this framework might want to conduct adversarial attacks against their own models to avoid being attacked by other people with bad intentions.

## 8 REPRODUCIBILITY STATEMENT

We provide code in the supplementary materials, and additional experiment details in the appendix.

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

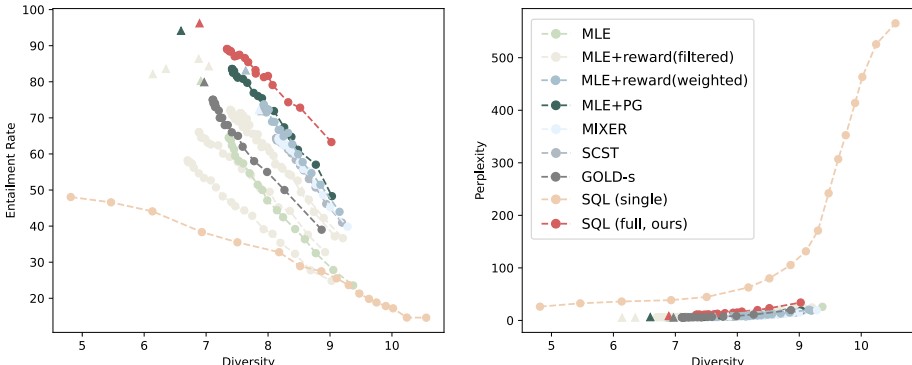

Figure 6: Entailment generation performance plotted against diversity (average of $H_1$ and $H_2$).

Adina Williams, Nikita Nangia, and Samuel Bowman. A broad-coverage challenge corpus for sentence understanding through inference. In *Proceedings of the 2018 Conference of the North American Chapter of the Association for Computational Linguistics: Human Language Technologies, Volume 1 (Long Papers)*, pp. 1112–1122, 2018.

Ronald J Williams. Simple statistical gradient-following algorithms for connectionist reinforcement learning. *Machine learning*, 8(3-4):229–256, 1992.

Lijun Wu, Fei Tian, Tao Qin, Jianhuang Lai, and Tie-Yan Liu. A study of reinforcement learning for neural machine translation. In *Proceedings of the 2018 Conference on Empirical Methods in Natural Language Processing*, pp. 3612–3621, 2018.

Yonghui Wu, Mike Schuster, Zhifeng Chen, Quoc V Le, Mohammad Norouzi, Wolfgang Macherey, Maxim Krikun, Yuan Cao, Qin Gao, Klaus Macherey, et al. Google's neural machine translation system: Bridging the gap between human and machine translation. *arXiv preprint arXiv:1609.08144*, 2016.

Wenpeng Yin, Jamaal Hay, and Dan Roth. Benchmarking zero-shot text classification: Datasets, evaluation and entailment approach. In *Proceedings of the 2019 Conference on Empirical Methods in Natural Language Processing and the 9th International Joint Conference on Natural Language Processing (EMNLP-IJCNLP)*, pp. 3905–3914, 2019.

Jingqing Zhang, Yao Zhao, Mohammad Saleh, and Peter J. Liu. Pegasus: Pre-training with extracted gap-sentences for abstractive summarization, 2019.

Ruiqi Zhong, Kristy Lee, Zheng Zhang, and Dan Klein. Meta-tuning language models to answer prompts better. *arXiv preprint arXiv:2104.04670*, 2021.

Li Zhou, Kevin Small, Oleg Rokhlenko, and Charles Elkan. End-to-end offline goal-oriented dialog policy learning via policy gradient. *arXiv preprint arXiv:1712.02838*, 2017.

Brian D Ziebart. Modeling purposeful adaptive behavior with the principle of maximum causal entropy. 2010.

Brian D Ziebart, Andrew L Maas, J Andrew Bagnell, and Anind K Dey. Maximum entropy inverse reinforcement learning. In *Aaai*, volume 8, pp. 1433–1438. Chicago, IL, USA, 2008.

# A APPENDIX

## A.1 APPLICATIONS AND EXPERIMENTS

### A.1.1 LEARNING FROM NOISY (NEGATIVE) TEXT

Please see Table 5 for beam search results, Figure 6 for additional results for `MLE+reward`, and Table 7 for examples.

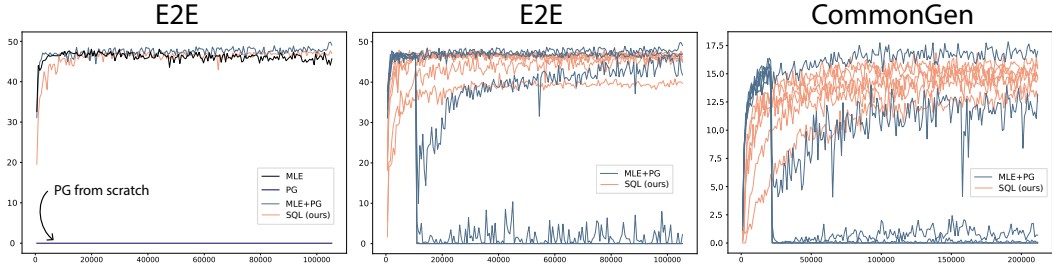

Figure 7: Training curves on validation sets. **Left:** Training curves on E2E with best hyperparameter configurations. **Middle:** Training curves on E2E with varying reward scale. **Right:** Training curves on CommonGen with varying reward scale.

### A.1.2 *Universal* ADVERSARIAL ATTACKS

Please see Table 4 for examples.

### A.1.3 PROMPT GENERATION FOR CONTROLLING PRETRAINED LANGUAGE MODELS

Please see Table 6 for detailed results breakdown, and Table 8-11 for examples. Examples are in the format: topic: [prompt] input sentence generated text.

### A.1.4 SUPERVISED TEXT GENERATION TASKS

Finally, we conduct experiment on standard generation tasks where clean supervised data is available. The study is to examine the capabilities of the proposed RL method to train a text generation model *from scratch*, which has been considered as exceedingly challenging for previous RL algorithms.

**Setup.** We study on two tasks, E2E (Novikova et al., 2017) and CommonGEN (Lin et al., 2020), and use the respective datasets pre-processed by (Gehrmann et al., 2021) which allow sequence-to-sequence modeling with standard transformers. We run four sets of methods: the standard MLE training (MLE); PG training from scratch (PG); joint MLE and PG training, with MLE initialization (MLE+PG); and our SQL training from scratch with both off-policy and on-policy updates (SQL). We use the standard BLEU as reward. We additionally investigate the training stability and sensitivity w.r.t hyperparameters, in particular the scale of reward. To this end, for MLE+PG and SQL, we vary the reward scale in $\{1, 10, 50, 100, 500, 1000\}$ and evaluate the respective performance under different scales.

| Model | MLE | PG | MLE+PG | SQL (ours) |
|---|---|---|---|---|
| **val** | 45.67 | 0.00 | 49.08 | 47.04 |
| **test** | 41.75 | 0.00 | 42.26 | 41.70 |

Table 3: BLEU results on the E2E val/test sets.

**Results.** Table 3 shows the performance on E2E of different models whose hyperparameters are picked using the validation set. We can see the proposed SQL that trains models from scratch achieves competitive results with the common MLE and MLE+PG. In contrast, the PG algorithm alone without MLE fails the training. Figure 7 (left) shows the respective training curves (on the validation set), demonstrating that SQL converges in an efficient and stable way as MLE.

We further demonstrate the sensitive of MLE+PG and SQL w.r.t the reward scale as a key hyper-parameter. Figure 7 (middle and right) shows the training curves of the two methods with varying reward scales. We can see SQL is significantly more robust as reward scale changes, while MLE+PG tends to collapse with improper reward scale configurations.

## A.2 Setup Details

Our evaluation follows the GEM Benchmark (Gehrmann et al., 2021) when applicable,[5] and otherwise same with the reward function used in training. We use a transformer model (Vaswani et al., 2017) based on Texar-Pytorch (Hu et al., 2019) by default, with $64$ hidden dimension, 3 blocks, and $4$ heads. For experiments that involve policy gradient training, we initialize the model with maximum likelihood training by default unless specified otherwise. We train soft $Q$-learning model from scratch with both off-policy (using data) and on-policy (using samples) by default except in §4.1 and §4.3, in which we find it beneficial to warm-up the model with just off-policy training. We apply similar tuning budgets to both soft $Q$-learning model, and policy-gradient (mostly the reward scale and top-$k$), based on performance on the validation dataset and sample qualities.

**Reward Functions** We use the robust entailment classifier (Nie et al., 2020) in §4.1,[6] one of the most used entailment classifiers on HuggingFaceHub in §4.2[7] and a zero-shot classifier based on BART (Lewis et al., 2020) to compute the topic score in §4.3.[8] To compute perplexities, we use a GPT-2 model (Radford et al., 2019) fine-tuned on the corresponding datasets for computing perplexity in §4.1 and 4.2, and a distilled GPT-2 model in §4.3 without fine-tuning.[9] We simply set reward weights to $1.0$, except in §4.2, where we changed the entailment weight to $0.5$, log-likelihood and repetition penalty weight to $5.0$.

### A.2.1 Setup Details: §4.1

We study using the SNLI dataset (Bowman et al., 2015), a dataset commonly used in training an entailment classifier. The original dataset contains *(premise, hypothesis)* sentence pairs, where the hypothesis may or may not entail the premise. We sub-sampled $50,000$ training examples from the corpus such that the hypotheses have an average entailment probability of only $50\%$ in terms of the premises, and over $2/5$ examples have entailment probabilities less than $20\%$, which can be seen as negative (contradictive) examples. The resulting training set poses a significant challenge for the models to learn from the noises.

The RL algorithms (including PG and ours) permit us to plug in arbitrary reward functions to drive learning. Based on the goal of the task, we use the following intuitive rewards to ensure entailment accuracy and language quality: (1) a robust entailment classifier (Nie et al., 2020) that measures the entailment score of a generation in terms of the input premise, (2) a GPT-2 language model (Radford et al., 2019) that measures the log-likelihood of the generation as an indicator of language quality, and (3) BLEU score w.r.t the input premises as another language quality reward that avoids trivial outputs. We sum together all rewards with weights $1.0$.

### A.2.2 Setup Details: §4.2

We study the task of attacking an entailment classifier. In particular, we aim to attack one of the most popular entailment classifiers on HuggingFaceHub.[10] The attack generation model generates adversarial text without conditioning on any inputs so that the generated attacks are universal to all premises. The generation model is trained with mostly the same setting as in §4.1, where the entailment classifier to be

| Model | Generation | Rate |
|---|---|---|
| MLE+PG | it 's . | 90.48 |
| SQL (ours) | the person saint-pierre-et-saint-paul is saint-pierre-et-saint-paul . | 97.40 |

Table 4: Entailment attack samples and respective entailment rates across all test premises. For example, the adversarial sample by SQL is considered to entail 97.40% test premises by the entailment classifier.

---

[5] https://github.com/GEM-benchmark/GEM-metrics

[6] https://huggingface.co/ynie/roberta-large-snli_mnli_fever_anli_R1_R2_R3-nli

[7] https://github.com/pytorch/fairseq/tree/master/examples/roberta. This classifier is **ranked #1** (as of May 20, 2021) based on https://huggingface.co/models?search=nli.

[8] https://huggingface.co/facebook/bart-large-mnli

[9] https://huggingface.co/distilgpt2

[10] https://github.com/pytorch/fairseq/tree/master/examples/roberta, which is ranked #1 as of May 20, 2021 based on https://huggingface.co/models?search=nli.

attacked is used as entailment score reward functions. Besides, we additionally include a token-level repetition penalty reward, which empirically benefits readability. Finally, we use the MultiNLI dataset (Williams et al., 2018) which includes more diverse examples than the SNLI used above.

We compare our `SQL` with `MLE+PG`. We use all hypotheses in the MultiNLI dataset as the training data for the MLE training in `MLE+PG` and the off-policy updates for our `SQL`. We do not compare with previous specialized adversarial text attack methods, because they either are not applicable to the universal attack setting (Morris et al., 2020; Jin et al., 2020; Ebrahimi et al., 2017), or were not designed to generate human-readable sentences (Wallace et al., 2019). Besides, it is worth noting that the general RL algorithms have an additional advantage of doing *black-box* attacks. That is, the algorithms only require the ability to query the entailment classifier for entailment probability, without need of knowing the internal structure of the classifier (e.g., for computing gradients) as in previous attack algorithms (Ebrahimi et al., 2017; Wallace et al., 2019).

For top-$p$ sampling results, we sample a hypothesis for each premise and measure the average attack rate across the dataset. This is because sampling multiple hypotheses, each for all premises, and measure performance are expensive. Since the hypotheses are sampled input-independently, this should be a good approximation.

### A.2.3 SETUP DETAILS: §4.3

Following (Dathathri et al., 2019), we aim to control the generation to have one of 7 topics (e.g., "science"); the generated prompt is prepended to one of 20 input sentences (Figure 4) for the pre-trained LM to generate continuation sentences. There is no direct supervision data available for training the prompt generator. We randomly create some noisy text as the training data for MLE baselines below and for off-policy updates for our algorithm. Specifically, the noisy text is created by sampling keywords and topics from the list used in (Dathathri et al., 2020) and a paraphrase generation model.

Figure 4 shows the architecture of prompt-based controllable generation. We compare our `SQL` method with `MLE+PG` as before. At training time, for each generated prompt sample, the pretrained LM generates 2 continuation sentences for evaluating average reward. We use a *zero-shot* classifier to evaluate the topic accuracy of the continuation sentences. That is, we do not assume access to classifiers pretrained on topic-specific sentences, because generating such topic-specific sentences is the goal of the task in the first place. We additionally use an LM to evaluate the log-likelihood of continuation sentences for measuring language quality. Since the prompt length could impact the generated sentences, we conducted experiments with maximum prompt length 5, 10, and 15. As ablation study, we also evaluate the SQL algorithm with only off-policy updates (i.e., without on-policy exploration), denoted as `SQL(off)`, and compare it with vanilla `MLE` training. At test time, given a topic, the trained prompt generator produces one prompt using beam search decoding. For each generated prompt, the pretrained LM generates 100 sentences using top-$k$ decoding (with $k = 50$) for evaluation. Finally, we also compare with two specialized controllable generation techniques based on pretrained LMs, namely `PPLM` (Dathathri et al., 2019) and `GeDi` (Krause et al., 2020), following similar procedures using their open-sourced code. We use a distilled GPT-2 model[11] as the pretrained LM to be controlled. We use the paraphrase generation model based on Zhang et al. (2019).[12] During decoding, we include `no_repeat_ngram_size= 2`, which improves readability.[13]

### A.3 THE SOFT $Q$-LEARNING FRAMEWORK

### A.3.1 COMPARISON WITH MLE OBJECTIVE

It is interesting to take a closer look at the above objective and compare with the common MLE training. Specifically, we notice the relations between the optimal $Q^*$, $V^*$, and $A^*$ functions: $A^*(\boldsymbol{s}_t, a_t) = Q^*(\boldsymbol{s}_t, a_t) - V^*(\boldsymbol{s}_t) = r_t + \gamma V^*(\boldsymbol{s}_{t+1}) - V^*(\boldsymbol{s}_t)$, where the first equation is the definition of $A^*$ (see Eq.7) and the second equation is due to Eqs.(12) and (7). We thus can see the regression target in the above objective as an approximation to the advantage function:

---

[11]`https://huggingface.co/distilgpt2`

[12]`https://huggingface.co/tuner007/pegasus_paraphrase`

[13]`https://huggingface.co/blog/how-to-generate`

| Model | Entl. Prob ↑ | Entl. Rate ↑ | PPL ↓ | $H_1$ ↑ | $H_2$ ↑ |
|---|---|---|---|---|---|
| MLE | 75.62/75.86 | 79.75/80.23 | 5.49/5.45 | 5.46/5.42 | 8.47/8.40 |
| GOLD-s (Pang & He, 2021) | 74.55/76.03 | 78.69/79.89 | 5.55/5.50 | 5.50/5.49 | 8.48/8.45 |
| MLE+PG | 90.16/89.73 | 95.18/94.13 | 6.38/6.31 | 5.23/5.20 | 8.02/7.99 |
| SQL | 91.94/91.55 | 96.26/96.21 | 8.41/8.42 | 5.59/5.58 | 8.20/8.21 |
| SQL (single)$^\dagger$ | 89.90/89.92 | 94.94/94.82 | 214.42/214.42 | 0.00/0.00 | 0.00/0.00 |

Table 5: Beam search results on entailment generation, in the format **val/test**. ↑/↓ indicates higher/lower is better. $^\dagger$SQL (single) achieves zero in $H_1/H_2$ as it generates a single token.

$\tilde{A}_{\bar{\theta}}\left(\boldsymbol{s}_t, a_t\right) := -V_{\bar{\theta}}\left(\boldsymbol{s}_t\right) + \gamma V_{\bar{\theta}}\left(\boldsymbol{s}_{t+1}\right) + r_t$. Therefore, by optimizing the regression objective, $\log \pi_\theta(a_t|\boldsymbol{s}_t)$, which is the log probability of generating token $a_t$ given preceding tokens $\boldsymbol{s}_t$, is encouraged to match the approximate advantage value $\tilde{A}_{\bar{\theta}}\left(\boldsymbol{s}_t, a_t\right)$, no more and no less. This is different from the objective of MLE where the model is trained to (blindly) increase the probability of the observed token $a_t$ given $\boldsymbol{s}_t$ and decrease the probability of the rest.

### A.3.2 VANILLA TRAINING WITH TEMPORAL CONSISTENCY

Much like the Bellman temporal consistency in standard $Q$-learning (Eq.3), in SQL, the optimal action-value function follows the *softmax* form of the temporal consistency (Ziebart et al., 2008; Ziebart, 2010; Fox et al., 2016; Nachum et al., 2017):

$$Q^*\left(\boldsymbol{s}_t, a_t\right) = r_t + \gamma \log \sum\nolimits_{a_{t+1}} \exp Q^*\left(\boldsymbol{s}_{t+1}, a_{t+1}\right). \tag{12}$$

We thus can derive a regression objective similar to the standard $Q$-learning (Eq.4):

$$\mathcal{L}_{\text{SQL, vanilla}}(\boldsymbol{\theta}) = \mathbb{E}_{\pi'}\left[0.5 \cdot \left(r_t + \gamma \log \sum\nolimits_{a_{t+1}} \exp Q_{\bar{\theta}}\left(\boldsymbol{s}_{t+1}, a_{t+1}\right) - Q_\theta\left(\boldsymbol{s}_t, a_t\right)\right)^2\right]. \tag{13}$$

Recall that $\pi'$ is an arbitrary behavior policy (e.g., data distribution), and $Q_{\bar{\theta}}$ is the target $Q$-network which is a slow copy of the $Q_\theta$ to be learned and is held fixed during the gradient updates. However, the above objective is inefficient due to exact the same reasons as in standard $Q$-learning discussed earlier, namely the unstable per-step bootstrapping-style training with sparse reward signals, plus the slow updates w.r.t only one token $a_t$ out of the large vocabulary (action space).

| Length | Model | legal | politics | computers | space | religion | science | military | Average |
|---|---|---|---|---|---|---|---|---|---|
| | | | | | Topic Scores | | | | |
| / | PPLM | 16.52 | 25.09 | 13.35 | 26.23 | 5.39 | 38.87 | 19.33 | **20.68** |
| / | GeDi | 40.51 | 83.40 | 9.32 | 70.90 | 18.69 | 12.46 | 86.40 | **45.96** |
| 5 | MLE | 17.28 | 13.44 | 7.26 | 42.27 | 45.24 | 39.31 | 63.75 | **32.65** |
| 5 | SQL (off) | 23.79 | 61.11 | 24.07 | 7.91 | 61.77 | 64.67 | 67.83 | **44.45** |
| 5 | MLE+PG | 29.45 | 74.16 | 72.49 | 57.39 | 65.62 | 74.31 | 76.86 | **64.33** |
| 5 | SQL | 11.79 | 70.57 | 66.37 | 58.80 | 65.60 | 69.24 | 83.15 | **60.79** |
| 10 | MLE+PG | 17.72 | 75.29 | 71.01 | 73.92 | 58.29 | 80.85 | 80.84 | **65.42** |
| 10 | SQL | 29.62 | 86.58 | 75.72 | 58.38 | 71.29 | 81.05 | 91.40 | **70.58** |
| 15 | MLE+PG | 40.18 | 81.47 | 47.14 | 82.64 | 76.21 | 84.82 | 89.31 | **71.68** |
| 15 | SQL | 48.08 | 77.94 | 70.04 | 87.43 | 75.46 | 85.94 | 77.36 | **74.61** |
| | | | | | Perplexity | | | | |
| / | PPLM | 13.52 | 12.81 | 12.79 | 13.56 | 12.98 | 12.43 | 13.38 | **13.07** |
| / | GeDi | 204.44 | 80.01 | 132.82 | 116.94 | 132.19 | 90.00 | 110.77 | **123.88** |
| 5 | MLE | 24.52 | 25.05 | 23.79 | 26.26 | 26.07 | 25.63 | 28.56 | **25.70** |
| 5 | SQL (off) | 25.48 | 22.70 | 25.10 | 26.64 | 25.84 | 27.45 | 27.19 | **25.77** |
| 5 | MLE+PG | 24.42 | 22.60 | 27.74 | 23.17 | 25.38 | 24.84 | 30.50 | **25.52** |
| 5 | SQL | 25.31 | 24.15 | 26.40 | 24.31 | 27.02 | 25.73 | 28.67 | **25.94** |
| 10 | MLE+PG | 28.25 | 23.49 | 27.82 | 26.88 | 31.62 | 25.31 | 33.74 | **28.16** |
| 10 | SQL | 25.23 | 25.37 | 26.20 | 26.97 | 25.02 | 27.11 | 32.76 | **26.95** |
| 15 | MLE+PG | 28.38 | 28.24 | 28.16 | 27.21 | 26.43 | 29.99 | 32.54 | **28.71** |
| 15 | SQL | 35.16 | 27.72 | 29.70 | 31.89 | 24.04 | 28.46 | 26.74 | **29.10** |

Table 6: Prompt generation results. Note that some of the numbers from GeDi are low because the topics are tokenized into two subword tokens, which the model was not trained with.

| | |
|---|---|
| Input: two men on bicycles competing in a race . 
 Generated: two men are riding bikes . | |
| Input: families waiting in line at an amusement park for their turn to ride . 
 Generated: families at a amusement park . | |
| Input: man in a black suit , white shirt and black bowtie playing an instrument with the rest of his symphony surrounding him . 
 Generated: a man is playing music . | |
| Input: a white dog with long hair jumps to catch a red and green toy . 
 Generated: a dog is jumping | |
| Input: a man in a black shirt is playing golf outside . 
 Generated: a man is playing golf | |
| Input: a man wearing sunglasses is sitting on the steps outside , reading a magazine . 
 Generated: a man is sitting outside . | |
| Input: a young child is jumping into the arms of a woman wearing a black swimming suit while in a pool . 
 Generated: a child is jumping into a pool . | |
| Input: a carefully balanced male stands on one foot near a clean ocean beach area . 
 Generated: a man is on the beach . | |
| Input: two young boys in swimming trunks are in the water looking at each other . 
 Generated: two boys are in the water . | |
| Input: a woman riding a bicycle past a car and a group of people on a sidewalk . 
 Generated: a woman rides a bicycle . | |
| Input: a female sings and plays into a microphone and a male in green striped shorts plays a hand drum 
 Generated: a woman is playing music . | |
| Input: a little girl wearing a cardboard diner hat is finishing off some onion rings at a restaurant . 
 Generated: a little girl is at a restaurant . | |
| Input: young woman celebrates getting a strike during a bowling game . 
 Generated: a young woman is present | |
| Input: black greyhound dog racing down a track . 
 Generated: a dog runs down a track . | |
| Input: several men on stage having a discussion . 
 Generated: men are talking on stage . | |

Table 7: Entailment generation samples from SQL (beam search, validation dataset).

legal: legal space religion and space In summary, a good understanding of these concepts is that by giving an explicit understanding to a person, they provide an avenue to be studied and studied. But the concept of one person being a space is also very confusing, and can be very difficult to obtain.\n So,

politics: the primary referendum is In summary, the outcome will be a referendum on the EU membership for the first time of its kind for EU citizens, and the full extent of the benefits of a single market and a flexible single EU state."

computers: macintoshintoshintoshintosh In summary, it appears that the company and IBM products are currently in need of upgrading the computer. This can be seen in a detailed review of the Macintosh version of Windows Vista and XP. However, when looking at the changes made by the HP Macintosh hardware and software versions of

space: legal space science and space In summary:\n\n The purpose of this paper is to investigate and test the theory of space space and other objects. This project will support NASA.s efforts to demonstrate these theories, and to propose other relevant new theories.\n This paper provides the following arguments for the

religion: space legal religion religion religion In summary, to the author the current discussion is the position of the Church and the community. While we acknowledge that we should not be commenting upon claims such as our recent cases or the other ones that contradict our view, we conclude it is appropriate to include these cases. Further

science: the chemical microscope is In summary, the most sophisticated of these experiments is a technique that gives no obvious, no apparent way of revealing that the material was obtained. In this study, we examine how the compounds in the samples in question make up the composition of the chemical and its properties. The chemical composition

military: arms defense battalion battalion cavalry In summary: 6th Panzer Field Division, Second Division.\n\n The main task of the battalion in the main counterinsurgency campaign was to counter the enemy in any counter-incursion. The main objective of this campaign is to eliminate enemy groups and the remnants of

legal: legal space religion and space This essay discusses the idea of space and time as a space, in both theoretical and conceptual terms, as not an individual time period or anything else. The emphasis is on time itself, rather than having a fixed central space. Space was the object of the first chapter, and

politics: the primary referendum is This essay discusses the nature of the EU referendum. The purpose of this essay is to shed light on the importance of a public referendum, on a question of whether the decision of an EU member states to remain in the European Union is constitutional and thus in accord with constitutional guarantees of sovereignty

computers: macintoshintoshintoshintosh This essay discusses hardware devices and software systems for Mac OS X, MacOS X and Linux. To view the latest version of Macintosh OS: Mac 8.7.x\n\n For more information or for information about Macintosh systems, visit Mac MacSystems.\n More

space: legal space science and space This essay discusses science for teens, adults and teenagers.\n\n When the idea of studying space was first implemented as a method to test, the question was: What if a student has been "comfortable" with space without its body? What would their body like to be

religion: space legal religion religion religion This essay discusses an alternative religion that focuses on the role of a particular religion and views some form of religious ethics as the form when the law is applied to that particular religious community . This discussion is concerned with the status of faith for individuals or groups which may be members and members

science: the chemical microscope is This essay discusses the mechanisms of reaction with a focus on the molecular structure of nucleite and of enzymes within the cytoskeleton, thus making it easier to understand the process of metabolism and other elements of cellular life. In this essay, we use techniques such as the photochemical transfer

military: arms defense battalion battalion cavalry This essay discusses three main themes:\n\n 1) Lack of uniformed soldiers is an unacceptable and unconscionable strategy for the Army.\n 2) Poor and inadequate training does not compensate the soldiers, and may deprive them of the necessary and competitive training from their instructors

Table 8: Prompt samples from SQL.

legal: there liable injunction In summary, the court decided that defendants had a right to petition the high court to intervene, and therefore granted the injunction. However, a Court of Appeal decision in the US District Court ruled with no effect on the petitioned by the plaintiffs, citing two reasons.\n\n The US

politics: the primary referendum was In summary, the majority of Russians would support the idea of banning all human rights, and that would be a major part of the government's effort to build a new liberal economy, which President Vladimir Putin says would boost Russian tourism.\n\n The results were published in the

computers: the macintoshintosh In summary, the kernel has the best quality of life, and kernel security is at the heart of any OS development. The first time Linux released was when i386 released the Macintosh version, which had a rather low standard. Other issues like the introduction of the OS6 operating system

space: comet asteroid spaceship In summary, this asteroid asteroid ship is about to make its maiden flight to explore deep-space.\n\n "The asteroid craft was a very early attempt at the discovery of a new home for the world's first black hole," NASA said in a news release. "Today we

religion: faith faith faith salvation In summary, Christian beliefs are not a new way to use the time spent thinking about God's world as a source for faith. Faith is an effort to think of the world without fear that it might become a dangerous place for the human family. Because it represents the very essence that

science: climate research chemistry In summary of the study, this review aims to determine how in a single study where the same number of data was analysed, a new methodology is needed to better understand who produced a different graph than the one suggested. The paper will be published in issue #5, Issue #18.

military: the cavalry battalion a In summary, the army are a unit of the same type and in all, so there is no need to declare one. The unit does not constitute a cavalry unit or for use on troops.\n\n The army is not under the command of a brigade from the front. For

legal: there liable injunction This essay discusses the potential legal consequences of a stay in the United States for an indefinite period of time if the government continues to delay the process of de-instituting it. To apply such a request, all applicable laws shall apply either the same terms as the existing statutes. In

politics: the primary referendum was This essay discusses the electoral strategy against a candidate for governor of the Commonwealth.\n\n The survey of British voters in this survey provides an overview of what the candidates for the United Kingdom will be seeking in the next Parliament. In the general election a few seats will lead up to a

computers: the macintoshintosh This essay discusses the various problems of the Macintosh, the first two-year running environment. An early version of this paper was originally published in 1982. The MacSX was not designed and managed by Kia.\n\n Macintosh\n The mac has been a family invention

space: comet asteroid spaceship This essay discusses a topic: the impact of two of the Earth's two-thirds comet-sized moon Charon on Earth, and why asteroids are so close to the sun; why people are looking for ways to find a way to keep Earth-shaped asteroids out of orbit.

religion: faith faith faith salvation This essay discusses the impact religion has on the American experience and in American culture. Since the beginning of my career I have found that faith and belief have often been linked to economic growth, social development and education. I believe that all people need to know that there is no reason for

science: climate research chemistry This essay discusses the role of molecular information and its inter-action with the general organism and human health.\n\n "The idea of biological information is not really a new concept. We used genetic information as a medium to define, identify, and store information about biology and biology," explains Dr.

military: the cavalry battalion a This essay discusses the potential for the development of a small infantry brigade as an infantry regiment. It is also a contribution to the larger cavalry corps as it would require a larger brigade for battle. For more information see the original article on this page.

Table 9: Prompt samples from MLE+PG.

legal: In summary Currently: In 1966 the Act was amended into state of law through amendments.\n\n\n Defent No. 1 etc 695 [The character in question for judicial decision purposes; participation t concerned you; "but not acceptance.")\n\n Generally held: Just

politics: In summary Senate candidates, senator (Republican); senator (Democrat); and opinion-former (2002-08). - 2012 Senate results are based on the federal Election Commission's October 2016 Current Opinion Polling Reports. Key figures : Open Gallup poll Most Americans view the

computers: In summary: 12-16 add-on chips. Trace out the type predefined ORDER parameters, and write to /dev/tty with them.\n\n\n\n\n\n\n\n Roundset sizes with mm(831x810 x870 x81f);

space: In summary Space Station - Farm Station (1985 by Mike Lazarra) Here is an article developed by Maregnus Spirit Experimentator on WinViotrv - An exploration benefit for compute-enriched array data densities (UPERS).This thesis home

religion: In summary nice things about Android 6.1 Jelly Bean!\n Searching for OP lag fixes one of my cllcs or some other improvements that's fixing a bug due to this nerf! (model causing Huge Frame Decay!) It also fixed an upper turret hook

science: In summary Computer Age Experience Overview\n\n\n\n Networking skills are the most developed skill set for Internetthumb members at universities at this time. In computer science, we are introducing various gatekeepers to intellectual property ownership and cyberware acquisitions, entry program makers post a

military: In summary Army Sgt. Harold Tolbard (5/16/2018) Lt. Gen. Michael Bachaes 1 Dickie Powell 2 Lt. Zachary Bram 9 *Gen. Robert Eisen: Warfighter – Soldier + Genoured\n\n\n – Senior Bush Doctrine

legal: This essay discusses Illinois cases on issues such as drug trafficking and drug Social Security.

politics: This essay discusses federal ethics as the key area on which current and past state and local governments have been operating.

computers: This essay discusses the very development of alternative technology for young people.

space: This essay discusses NASA's StarHubble satellite mission development. Transcript here.

religion: This essay discusses various aspects of the relays of mediocality and Hammazanna.

science: This essay discusses Linux desktop computing, and IRI video-game applications.\n\n The zooming in — even after the GNOME 3 transition came to an end, is all about figuring out how you have run a software operating system so vital that any hacker can mine it

military: This essay discusses military courage that included in the combat operations in Iraq and Afghanistan.

Table 10: Prompt samples from GeDi.

legal: In summary we have published in the journal Nature Neuroscience: A systematic review of human brain tissue has found no evidence for any association between the presence of the presence of a particular form of human neuropathy in the brain, a condition that is not normally associated with cognitive impairment. We found that

politics: In summary we have a list of 10 of the best and most common types of drugs for people with HIV. This is a very short list of recommendations from a national and international community.\n\n\n\n This article has been updated to make the official state of the EU state of

computers: In summary, we believe that the current system has no way of doing anything about it.\n\n\n\n The following steps are taken to get the system working.\n\n 1. Install a new operating system with a Linux Mint operating system\n 2. Start a new Linux Mint operating

space: In summary we have some important news from the moment of the year and some important information about these two major planets. This new discovery is the first to confirm this important planet has an active life in its home planet, a planet with a mass of about 5.8 billion tons. It

religion: In summary, we believe that the current administration has no way of doing anything about the Benghazi attacks. This is a very interesting story, and I think it has been a very nice surprise. This is a very nice and well thought out piece that is a must for the

science: In summary we use this approach to evaluate if the number of data points (in the dataset) that are relevant for each data set is the same (in this case, the data are not in one data set). In this approach we can test the data points in a different way.

military: In summary we have some important news from the moment of the year and some important information from the moment of the year.\n\n\n\n\n We've also added an additional update for our new feature, which includes:\n • Improved access and access in all of the main

legal: This essay discusses how you can build a community of dedicated people. If you're a member of a community of people who want to contribute to the environment, you'll also be helping them build communities in order to support the local economy, and the future of the city. The latest report

politics: This essay discusses how we can build on previous research findings about the role religion plays in human development in human development. This is a very interesting and highly entertaining story. What is an "independent" political party in the United States, the U.S. political party, and the United

computers: This essay discusses how you can build a new browser to view and share your favorite web sites.\n\n\n A browser that is open source can also be built from a web browser, which can be a browser that does not allow browser extensions (e.g. Firefox, Chrome, Opera

space: This essay discusses how you can build a life with a healthy diet and how you can use it when you're ready to move forward. It's a very simple approach to building a life with a healthy diet and what it means to be healthy and healthy for the

religion: This essay discusses how you can build a new game without having to play the original game, and how you can make a new title that is completely different to the original. It has been around since 2007, when the first game, The Elder Scrolls IV: Oblivion, was released in the PlayStation

science: This essay discusses how we can build on previous research findings about the role of obesity in human metabolism and how we can improve our health.\n\n\n\n In this essay, we explore why eating a whole whole diet does not help prevent obesity (1). We find that a whole food diet

military: This essay discusses how you can build a community with the help of friends and family.\n\n\n\n\n "The people around me are the ones who need help. They are the ones who need help. They are the ones who are not alone."\n - Michael\n "It's

Table 11: Prompt samples from PPLM.

