# OpenReview forum: "Text Generation with Efficient (Soft) $Q$-Learning"
_ICLR.cc/2022/Conference — ICLR 2022 Submitted_

### Official Review · Reviewer_2uap · 2021-10-28

**Correctness:** 3
**Technical Novelty And Significance:** 2
**Empirical Novelty And Significance:** 2
**Recommendation:** 5
**Confidence:** 4

**Main Review:**

This paper models text generation as a reinforcement learning problem, and uses the existing reinforcement-learning techniques (soft Q-learning and path consistency learning) to relieve the sparsity reward and large action space problems in the existing reinforcement learning based text generation methods.

Strong Points:
- The paper is well-written.
- The illustration of this paper is clear and good.
- The proposed methods achieve moderate improvement over previous methods.


Weak Points:
- The novelty of this paper seems limited as the techniques used in this paper are directly from the reinforcement learning area (soft Q-learning and path consistency learning). Considering that the reinforcement learning methods had been widely used in the text generation methods, e.g, MIXER [1] SeqGAN [2], the idea of using reinforcement learning methods for the text generation seems less novel.

- One of the motivations of this paper is to relieve the sparsity problem in text generation. However, this problem had been clearly pointed by the previous text generation methods, e.g. LeakGAN [3] and IRLGAN [4]. Specifically, IRLGAN tried to relieve this problem using the inverse reinforcement learning method, which is highly related to the submission. However, the paper does not compare with them and even does not mention them. This is limiting.

- How significant is the sparsity problem in the chosen three text generation tasks? If the authors emphasise the sparsity reward problem is significant in the text generation, the related experiments, e.g. generating different lengths of texts, should be performed. However, the related analysis is absent in the paper.

- One of the baselines in this paper is MLE+PG. There are many MLE+PG text generation methods, but authors do not clearly state which one they use in the paper, which leads to confusion.

- The paper shows the generation results of the prompt generation which is an interesting application of text generation. However, there are many meaningless and less fluent generated prompts in the shown examples, e.g. macintoshintoshintoshintosh in Table 7. What causes this phenomenon?

- In addition to the prompts results, It would be better to show generated examples of other tasks.

[1] Ranzato, M. A., Chopra, S., Auli, M., & Zaremba, W. (2015). Sequence level training with recurrent neural networks. arXiv preprint arXiv:1511.06732.

[2] Yu, L., Zhang, W., Wang, J., & Yu, Y. (2017, February). Seqgan: Sequence generative adversarial nets with policy gradient. In Proceedings of the AAAI conference on artificial intelligence (Vol. 31, No. 1).

[3] Guo, J., Lu, S., Cai, H., Zhang, W., Yu, Y., & Wang, J. (2018, April). Long text generation via adversarial training with leaked information. In Proceedings of the AAAI Conference on Artificial Intelligence (Vol. 32, No. 1).

[4] Shi, Z., Chen, X., Qiu, X., & Huang, X. (2018, July). Toward diverse text generation with inverse reinforcement learning. In Proceedings of the 27th International Joint Conference on Artificial Intelligence (pp. 4361-4367).

**Summary Of The Paper:**

This paper proposes a new text generation framework based on the existing soft Q-learning of reinforcement learning. The experiments demonstrate that the proposed text generation method achieves superior performance to baselines.

**Summary Of The Review:**

In summary, this paper is well-written but it lacks relevant analysis and experimental evidence for the problem they want to solve. Further, the method they proposed is directly from the reinforcement learning area, which significantly reduces the contribution of this paper. Last, the paper lacks the experimental comparison with the related work aiming to solve the same problem.

---

> ### Author Response · Authors · 2021-11-20
> **Response**
>
> Thank you for the feedback!
>
> * **Novelty**
>
> Please see our general response above for the summary of technical novelty and empirical contributions of the paper. We agree that using RL for text generation is not new. Indeed, our Introduction (sec.1) and Background (sec.2) sections have provided detailed discussions of the previous RL methods for text generation, and pointed out their limitations that motivate our work. The main technical contribution of our work is the *new* RL formulation of text generation from the perspective of soft-Q learning, which *differs from* the existing policy-gradient and normal Q-learning formulations. The new formulation further enables the application of advanced techniques such as PCL, leading to the superior performance on multiple text generation tasks.
>
>
> * **LeakGAN and IRLGAN**
>
> Reward-sparsity is a *well-known* challenge in RL for text generation, and we’ve cited a bunch of related papers in both Introduction (sec.1) and Related Work (sec.5). We appreciate the reviewer for pointing out these missing references and have updated the paper to include those (**highlighted with red color**). We note that LeakGAN and IRLGAN are not suitable for the text generation applications studied in our paper, because they rely on expert distribution (e.g., clean supervised data) and MLE training in various ways, while we do **not** have such expert distribution in our tasks. For example, IRLGAN initializes the generator with MLE, and it learns the reward approximator from the expert distribution (again through MLE). Besides, as also mentioned in the paper, MLE warmup implicitly biases the model, hence limiting its effectiveness in many emerging applications like the ones explored in this work.
>
> * **Analysis of sparsity**
>
> Thanks for the question. We indeed have included ablation studies showing the importance of overcoming the sparsity issue. Specifically, in Figure 3 (left), we show the comparison between our full approach (“SQL (full)”, with both single- and multi-step PCL training) and a simplified variant that uses only single-step PCL training (“SQL(single)”). The superiority of “SQL (full)” highlights the benefits of having the multi-step objective (Eq.11) which is designed for alleviating the sparsity challenge.
>
>
>
> * **MLE+PG**
>
> At the top of page 7, we have described the MLE+PG method: *“We initialize the model with MLE training, then train the model with combined MLE and PG losses”*. We’ll make it clearer.
>
> * **Prompts**
>
> The end goal of this task is to generate sentences of desired attributes (controlled by the prompts). The language quality emphasis is on the *generated sentences*, rather than the prompts. Prompts themselves do not necessarily have to be human-readable, as also noted in previous works such as [1] and [2].
>
> * **Generated examples of other tasks**
>
> Thanks for the suggestion. Appendix A.1.2 (Table.4) has shown the generated examples for the task of adversarial attack. We also added the generated examples for the entailment generation task, to Appendix Table 7.
>
>
> [1] Universal Adversarial Triggers for Attacking and Analyzing NLP
>
> [2] Towards Controllable Biases in Language Generation

---

### Official Review · Reviewer_FPpH · 2021-11-03

**Correctness:** 3
**Technical Novelty And Significance:** 1
**Empirical Novelty And Significance:** 2
**Recommendation:** 5
**Confidence:** 3

**Main Review:**

1. I think when mentioning text generation, people will recall language modeling or machine translation. This paper only did experiments on some tasks which I don't think it's central to the topic. I think it's a misleading title and content. Since authors used this title, I think at least one of these 2 central tasks should be added in experiments.

2. I think overall all the technical details are not new (correct me if I am wrong, and in this case kindly pointed out what's the contribution). It reads to me that the major contribution is trying some existing technical methods in these sort of problems. If that's the case, I feel it's more appropriate to submit to EMNLP I guess.

3. On technical issues, I feel like the claim is not well justified. In many places, authors claims SQL is the important thing in overcoming the existing problems. However, in SQL the major concept to me is the entropy term. If that's the case, that means we want the term to be large and it translates to we want an evenly distributed actions space and in this thinking, normal SARSA or Q-learning or with larger epsilon greedy search should also work. I think if SQL is the reason, this should also be verified.

4. Somehow it reads to me that SQL itself is not even enough to overcome the challenges so path consistency learning is the most important. So I think 1) there should be an ablation analysis on SQL only, and 2) I feel path consistency learning can be applied to other RL problems as well, which should also be included. And if it cannot be combined, please discuss the reason.


**Summary Of The Paper:**

Authors proposed to use soft Q-learning (SQL) to formulate the RL problem in sequence generation. The stability can be increased when techniques of path consistency learning is corporated.

**Summary Of The Review:**

Overall, I have the experience in using RL for machine translation so I agree with the authors that those problems mentioned in the paper are indeed challenging. But I am not familiar with SQL but from the derivation provided in the paper I am skeptical on the effectiveness of SQL. Since most if not all technical stuff are existing work, I think the novelty is limited and thus justification of each component in the method is important to me.

---

> ### Author Response · Authors · 2021-11-20
> **Response**
>
> Thank you for the feedback.
>
> **1. Text generation tasks**
>
> We note that text generation is a rather broad area spanning a wide range of central tasks, such as (besides language modeling and MT) data-to-text generation, summarization, dialog, adversarial text, paraphrase, storytelling, controllable generation, prompt generation, text infilling, and so forth. All of the three tasks studied in the paper are central NLP/NLG tasks gaining significant traction in the area. For example, please see the latest survey paper on prompt generation [1], and one of the (many) paper lists related to text adversarial attacks [2], etc.
>
> [1] Pre-train, Prompt, and Predict: A Systematic Survey of Prompting Methods in Natural Language Processing
>
> [2] https://github.com/thunlp/TAADpapers
>
>
> **2. Technical novelty**
>
> Please see our general response about our technical novelty and empirical contributions.
>
>
> **3. SQL and entropy**
>
> As explicitly described in the paper (e.g., abstract, the second last paragraph in introduction, section.3.2), SQL serves as the basis for a new RL formulation of text generation that bridges the gap between advanced RL and the generation problem, which enables us to further draw from the latest RL techniques (e.g., path consistency learning) to overcome the challenges in text generation. Vanilla SQL alone with the additional entropy term is not sufficient (e.g., see the first paragraph of section.3.2, or Appendix A.3.2), and thus section.3.2 is devoted to introducing the extension, namely PCL, as the final solution.
>
> Similarly, normal Q-learning (even with larger epsilon greedy) will not work either, as discussed in detail in section 2.1 “Value-based RL”.
>
>
> **4. Path consistency learning and ablations**
>
> **1)** As clarified above, we do not consider vanilla SQL alone as an effective solution to text generation (just like previous normal Q-learning). Instead, vanilla SQL alone is the basis that enables us to further introduce its advanced extension, namely PCL, for the final effective solution.
>
> We’d also like to note that, in the experiments, we’ve included a few ablations to justify each key component in our algorithm. For example, in Figure 3 (left), we have the comparison between our full approach (“SQL (full)”, with both single- and multi-step PCL training) and a simplified variant that uses only single-step PCL training (“SQL(single)”). This comparison highlights the benefits of having the multi-step objective (Eq.11) for overcoming the sparse reward issue. Besides, in Figure 5 and Table 1, we have the comparison between our full approach (“SQL”, with both on- and off-policy learning), its simplified variant with only off-policy learning (“SQL (off)”), and the MLE baseline. The results demonstrate the advantage of combining on- and off-policy learning as enabled by our formulation in a principled way. The results also show that the off-policy component of SQL is better than standard MLE training, as it incorporates reward signals instead of just blindly following the (noisy) data.
>
> **2)** The paper specifically focuses on the text generation problem. Application of PCL to other RL problems is out of the scope of this paper.

---

### Official Review · Reviewer_jb7t · 2021-11-03

**Correctness:** 4
**Technical Novelty And Significance:** 3
**Empirical Novelty And Significance:** 3
**Recommendation:** 6
**Confidence:** 4

**Main Review:**

Strength: This paper establishes a connection between the text generation problem and path consistent learning. It leads to a principled and elegant RL-based text generation method that naturally inherits many desirable advantages from PCL algorithm, such as being capable of using both on- and off-policy samples. It is widely applicable to text generation tasks where MLE training is not directly applicable, and demonstrate superior performance on 3 such tasks.

Weakness: The work is almost a direct application of PCL to text generation, and there is no newly developed RL algorithm for text generation. This may not be a serious weakness given that this paper mainly focuses on RL-based text generation problem, and the good performance could be encouraging for research on RL-based text generation methods. The experiments on standard MLE-based tasks are relatively small scale. Demonstrating competitive performance of SQL on standard machine translation tasks would be more convincing.

Comments:
- It would be helpful to explain the intuition of the single-step PCL loss (9) and the multi-step PCL loss (10) in the text generation context. In particular, compare its intuition to the MLE loss could make it better received in NLP community.

- It seems that the perplexity of the proposed method (SQL full) is not as good as the others (Figure 3 Middle and also Table 1). The authors may need to discuss it more thoroughly.

- Compared to the original PCL learning, the SBEED algorithm [Dai et al 2018] provides a provably-stable algorithm for path consistency learning when there is nonlinear function approximation (e.g., by using deep neural networks to parameterize the Q-function, the state-value function and the policy here).

[Dai et al 2018]: “SBEED: Convergent reinforcement learning with nonlinear function approximation”, Proc. ICML 2018.


**Summary Of The Paper:**

This paper considers the problem of learning text generation models using reinforcement learning. The problem is challenging in that RL algorithm becomes inefficient or unstable when dealing with large action space and the sparse reward situations in text generation. To address these problems, this paper adapts the path consistency learning approach to the text generation setting. It allows to train the model with both on- and off-policy samples, bridge the sparse reward signal to directly supervise the Q-function learning, and makes efficient updates to Q-values by considering all candidate actions together. Experiments result show that the proposed approach are effective in solving a wide range of applications where MLE are not applicable, where it achieves better performance than previous RL-based methods as well as task-specific methods. In addition, on standard MLE-based tasks, the proposed approach could also achieve competitive performance when training the models from scratch.

**Summary Of The Review:**

This paper establishes a connection between text generation and path consistent learning, which leads to an elegant RL-based text generation algorithm that inherits many advantages from PCL algorithm. The performance of the algorithm is encouraging for RL-based text generation, although having more large-scale experiments would be more convincing.

---

> ### Author Response · Authors · 2021-11-20
> **Response**
>
> Thank you for appreciating that our method is principled and elegant, and the empirical performance is strong.
>
> * **Novelty**
>
> We’d like to highlight that the main technical novelty of this work is the new RL formulation of text generation based on soft-Q learning. It is this *non-trivial* new formulation that bridges the gap between the latest RL advances (originating from Robotics and Control) and text generation, making it possible to “directly apply” PCL to text generation. We consider the enabling of the “direct application” of advanced RL techniques as a major strength of the work as it leads to simple and elegant solutions.
>
> Please see our general response for more discussions about our technical and empirical contributions.
>
> * **Standard MLE-based tasks**
>
> Thanks for the suggestion. We put the study on standard MLE-based tasks in the Appendix and did not claim it as our contribution in the present work (we’ve instead focused on the various tasks where MLE is not applicable). The current results on the MLE-based datasets are promising, and we consider larger-scale studies (e.g., on MT as you suggested) as our future work.
>
> * **Intuition of PCL objectives for text generation**
>
> This is a good suggestion! In fact, we’ve already included an intuitive explanation and comparison with the MLE objective in the Appendix A.3.1. Briefly, MLE trains the model to (blindly) increase the probability of the observed tokens, while PCL encourages the (log) probability of the tokens to match the approximate *advantage* values. We’ll make it clearer in the main paper.
>
> * **Perplexity**
>
> We clarify that the perplexity of our *SQL (full)* is already low and very close to that of other methods (e.g., the red curve in Figure.3 Middle). The text samples generated by SQL (full) and other methods are almost indistinguishable in terms of fluency. For example, the perplexity of SQL (full) in Table.1 is only 25.77, similar to that of MLE (25.70) and many others, and the samples are mostly sufficiently fluent (as also indicated by the low perplexity values). We’ll make this clearer.
>
> * **SBEED algorithm**
>
> Thank you for the pointer. We agree that, based on our RL formulation, SBEED can potentially be applied to and improve text generation straightforwardly. As we noted above, facilitating such “direct applications” of advanced RL techniques to text generation is one of the major advantages of our proposed formulation. We hope this work could inspire follow-up research to bring more recent RL approaches into text generation (SBEED included).

---

### Author Response · Authors · 2021-11-20
**General response to all reviewers**

We thank all reviewers for their insightful and encouraging comments. We’re encouraged by the reviewer’s appreciation that **(1)** the problem we’re tackling (i.e., Reinforcement Learning for text generation) is challenging and significant (R1 and R2); **(2)** our proposed RL method is principled and elegant (R1), and **(3)** achieves superior performance on a wide range of tasks (R1 and R3); **(4)** the paper is well-written with clear illustration (R3).

We emphasize that the **main technical novelty** of the paper lies in the new RL formulation for text generation based on *Soft Q-Learning*, which differs from the previous RL formulations based on policy gradient and conventional Q-learning (see sec.2). It is the new formulation that allows us to seamlessly take advantage of the latest successful techniques from the RL literature (in particular the path consistency algorithm) to overcome the longstanding challenges (e.g., sparse reward) in text generation. We expect that the new RL formulation would open up more opportunities to easily import existing and future RL advances for better text generation.

Our **empirical contributions** include studies on a wide variety of text generation tasks (generating from noisy/negative data, adversarial text generation, prompt generation), showing that our *general* approach consistently improves over not only previous text RL algorithms, but also diverse *task-specialized* methods. We’d like to highlight that all of the tasks are gaining significant traction in text generation research (e.g., see the latest survey of prompt generation [1]).

[1] Pre-train, Prompt, and Predict: A Systematic Survey of Prompting Methods in Natural Language Processing

*(Note: R1 - Reviewer jb7t; R2 - Reviewer FPpH; R3 - Reviewer 2uap)*

---

### Author Response · Authors · 2021-12-02
**More Comments/Questions Are Welcome**

Dear Reviewers and ACs:

We’ve made responses to all questions/comments by the reviewers earlier. We’re happy to discuss more if you have more questions/comments on our work. Thanks!

---

### Public Comment · ~Saeed_Najafi1 · 2023-02-10
**Application in Prompt Generation**

This paper has been applied on prompt generation using RL.
The follow-up paper is published at EMNLP 2022.

https://aclanthology.org/2022.emnlp-main.222/

---

### Decision · Program_Chairs · 2022-01-20

**Decision:**

Reject

**Comment:**

This work proposes an approach to improve non-ML based methods of text generation. It reformulates the problem with the soft Q-learning approach from RL instead of standard hard RL formulations from previous text generation work. By doing this, the work allows application of path consistency learning. This is an elegant formulation. However, this reformulation into soft Q-learning appears quite straightforward and so the application of path consistency learning does not require much change to be used for text generation. This limits the novelty of the work. The experiments are also relatively small-scale and consists of some non-standard tasks such as prompt generation (which is typically evaluated indirectly, the response to the prompts rather than the prompt itself). As the reviewers mention, evaluating on more large-scale standard tasks such as summarisation or dialog would be more convincing. Finally the work lacks references to recent works in the field, such as LeakGAN.